# Alpha-Synuclein and LRRK2 in Synaptic Autophagy: Linking Early Dysfunction to Late-Stage Pathology in Parkinson’s Disease

**DOI:** 10.3390/cells9051115

**Published:** 2020-04-30

**Authors:** Giulia Lamonaca, Mattia Volta

**Affiliations:** Institute for Biomedicine, Eurac Research-Affiliated Institute of the University of Lübeck, 39100 Bolzano, Italy; giulia.lamonaca@eurac.edu

**Keywords:** LRRK2, autophagy, Parkinson’s disease, alpha-synuclein, synaptic transmission, neuropathology

## Abstract

The lack of effective disease-modifying strategies is the major unmet clinical need in Parkinson’s disease. Several experimental approaches have attempted to validate cellular targets and processes. Of these, autophagy has received considerable attention in the last 20 years due to its involvement in the clearance of pathologic protein aggregates and maintenance of neuronal homeostasis. However, this strategy mainly addresses a very late stage of the disease, when neuropathology and neurodegeneration have likely “tipped over the edge” and disease modification is extremely difficult. Very recently, autophagy has been demonstrated to modulate synaptic activity, a process distinct from its catabolic function. Abnormalities in synaptic transmission are an early event in neurodegeneration with Leucine-Rich Repeat Kinase 2 (LRRK2) and alpha-synuclein strongly implicated. In this review, we analyzed these processes separately and then discussed the unification of these biomolecular fields with the aim of reconstructing a potential “molecular timeline” of disease onset and progression. We postulate that the elucidation of these pathogenic mechanisms will form a critical basis for the design of novel, effective disease-modifying therapies that could be applied early in the disease process.

## 1. Introduction

Parkinson’s disease (PD) is the second most common neurodegenerative disorder after Alzheimer’s disease (AD) and the most prevalent synucleinopathy worldwide.

By 2030, with the progressive increase in life expectancy, it is projected that up to 9 million people in the world will suffer from this disease, highlighting the medical, social and economic challenge for our society [1]. 

The defining factor of PD is age, affecting about 1% of the global population at 65 years and over 4% of individuals at 85 [2]. The disorder manifests when ~70% of dopamine (DA) neurons in the substantia nigra pars compacta (SNpc) have degenerated, and is characterized by the presence of intracytoplasmic inclusions mainly composed of alpha-synuclein (aSyn) in the surviving neurons, named Lewy bodies and neurites [3]. SNpc cells project to the striatum forming the nigrostriatal pathway, which influences movement, gait, balance, posture control and action selection. Thus, their loss explains the cardinal motor symptoms of the disease: bradykinesia, rigidity and resting tremor. 

Clinical observations confirm that PD is not a single disease with a single pathogenesis but rather has a multifactorial etiology, with the interaction between genetic and environmental factors as primary influences. Although most of PD cases are idiopathic (i.e., of unknown etiology), about 10% of all cases are linked to a defined genetic cause and over 20 causative variants have been identified to date; in addition, recent advances in genome-wide association studies (GWAS) have uncovered several risk factors for idiopathic PD [4]. 

The genetic evidence suggests several biological processes may underlie the pathology, including oxidative stress, mitochondrial dysfunction, neuroinflammation, impaired intracellular protein catabolism, vesicle trafficking and synaptic dysfunction [5,6]. 

This complicated scenario of PD neurodegeneration increases the challenges in designing a disease-modifying therapy, which is currently lacking. Nowadays, only a few pharmacological options are available, and while effective in the short term, their effects are temporary and are limited to the control of motor symptoms. Thus, it is widely accepted that identifying pathogenic mechanisms underlying the disease would be required to develop therapeutic strategies able to halt or slow disease progression. 

Synaptic function requires a large number of proteins; thus, the accumulation of defective ones and the subsequent loss of protein homeostasis may be a leading cause for synaptic dysfunction. In this scenario, the autophagy-lysosome pathway (ALP; the system deputed to protein and organelle degradation) might contribute to maintenance of synaptic proteostasis. The breakdown of this process has been suggested to lead to synaptic failure, axon degeneration and ultimately to neuron loss [7,8]. 

Since the pathological processes in PD start decades before the occurrence of motor symptoms, it is vital to discern both the primary mechanisms involved, and to target them early in order to modify disease progression. 

Recently, it has been suggested that synaptic dysfunction might be an early event not only in PD, but common to several neurodegenerative diseases including, AD and Huntington’s disease (HD) [7,9]. How synaptic dysfunction disrupts neuronal circuitry and which molecular mechanisms cause synaptic defects in PD still remains to be fully elucidated, but recently accumulating evidence has connected synaptic dysfunction with ALP [10,11]. Since ALP has been mostly viewed as a modulator of proteinopathy, this angle provides a novel link between early and late processes in neurodegeneration.

Aging is associated with defects in protein quality, a phenomenon that is enhanced in age-related neurodegenerative conditions. In our opinion, studying how these mechanisms progress will provide a powerful tool to uncover the processes underlying disease onset. For this reason, in this review we will discuss how ALP and synaptic dysfunction may share common cellular processes centered on Leucine-Rich Repeat Kinase 2 (LRRK2) and aSyn, highlighting parallels in the mechanisms of PD pathogenesis and progression. A more detailed appreciation of these processes has the potential to uncover novel therapeutic targets with broad efficacy, and to suggest suitable intervention timelines for initiation of treatments.

## 2. Synaptic Dysfunction in Parkinson’s Disease 

Synapses are highly specialized units for neuronal communication, converting the electrical signal of an action potential into a chemical signal. 

When an action potential reaches a nerve terminal, it triggers a quantal release of neurotransmitters that propagate a neuronal signal to the post-synaptic terminal. The release of neurotransmitters involves the fusion of synaptic vesicles with the plasma membrane. All these processes rely on precise protein quality control systems, but in (postmitotic) neurons, defective proteins cannot be replaced by repeated cellular divisions. In addition, synapses are often far from the neuronal body, where protein synthesis primarily occurs. Taken collectively, these characteristics may increase the possibility that alterations in the synaptic protein turnover machinery can lead to synaptic dysfunction and neurodegeneration. 

In neurodegenerative diseases, synaptic loss and dysfunction generally occur before the soma is affected. 

In the classic toxin animal models of PD, such as rodents treated with 6-hydroxydopamine (6-OHDA) or 1-methyl-4-phenyl-1,2,3,6-tetrahydropyridine (MPTP), alterations in spines and dendrites have been correlated to DA depletion and are linked to dysfunctions in synaptic neurotransmission.

In post-mortem PD brains, dendritic spines of the medium spiny neurons (MSNs) are reduced in the striatum. Critically, this is replicated in 6-OHDA-treated mice indicating that loss of striatal DA is linked to morphological alterations of synaptic structures. It should be noted, however, that in these cases, synaptic alterations may appear to temporally follow neurodegeneration [12,13]. 

Dendritic spine density was also found to be significantly decreased in MPTP mice, where 30 days of treadmill running exercise contributed to recovering dendritic spine density and arborization in MSNs of both the direct and indirect pathway [14].

Similar findings have been made in both 6-OHDA rats and MPTP primates, in which the rate of dendritic spine loss correlates with the relative degree of striatal DA denervation [15,16].

Additionally, long-term potentiation (LTP) and long-term depression (LTD) are altered in MSNs from 6-OHDA animals, indicating that DA is required for the correct expression of these forms of synaptic plasticity and that synaptic function is affected in PD conditions [17,18].

Moreover, the accumulation of aberrant or misfolded protein aggregates that lead to a decrease in protein clearance by autophagic degradation is thought to be a key feature of several neurodegenerative disorders [19].

PD is characterized by the pathological neurotoxic aggregation and accumulation of aSyn, encoded by the *SNCA* gene [20]. Among other functions, aSyn is physiologically implicated in presynaptic neurotransmitter vesicle modulation [21]; indeed, aSyn is normally enriched at the presynaptic terminals, where recombinant wild-type (WT)-aSyn also localizes [22]. Recent studies suggest that aSyn impacts presynaptic function by regulating synaptic vesicle dynamics and pools. It has been demonstrated that aSyn mediates the assembly of the SNARE complex, which is required for vesicle release [23]. In addition, overexpression of WT- or A30P-aSyn in chromaffin and PC12 cells is associated with impairment of DA secretion, indicating that aSyn inhibits DA vesicle release by downregulating vesicle access to the presynaptic membrane. In DA neurons, aSyn suppresses tyrosine-hydroxylase (TH) activity, increases DA storage into vesicles and reduces the activity of the DA transporter (DAT) [24,25]. This evidence indicates that synaptic alterations might precede neuronal loss, moving the scope to earlier stages of disease.

Autophagy contributes to the degradation of aSyn providing another avenue through which it may impact synaptic function. Normally, WT-aSyn is degraded by chaperone-mediated autophagy (CMA), one of the three major types of autophagy (see below). Macroautophagy is proposed to compensate for impaired CMA [26], and defective autophagy enhances the deposition of aSyn aggregates in Lewy bodies [27,28]. Indeed, impaired autophagic clearance leads to deposition of aSyn in presynaptic terminals of mice lacking Atg7, supporting the hypothesis that synapses are a major target of aSyn pathology [29,30]. 

In addition, aSyn has been reported to be implicated in the modulation of post-synaptic functions, and not only in neurotransmitter release. Indeed, exogenously administered aSyn oligomers impair the localization of the NMDA-R subunit GluN2A at the post-synaptic membrane with downstream impact on synaptic plasticity in the striatum [31]. In addition, overexpression of mutant A53T-aSyn impairs AMPA-R signaling in the mouse hippocampus, producing deficits in LTP and memory. Interestingly, these defects appear in the absence of neurodegeneration [32], reinforcing the hypothesis that alterations at synapses occur before overt neuronal death. The synaptic function of aSyn has been postulated to be connected to its interaction with membranes, where oligomerization could mediate both physiological pre- and post-synaptic roles, but also be implicated in pathology [33]. 

In this regard, the structure of aSyn has also been studied in relation to its aggregation properties in the attempt of understanding the neuropathology. Three different domains are present in the protein: the N-terminal region, consisting of several repeats with a highly conserved motif (KTKEGV); the central domain, known as the non-amyloid component region (NAC), which is predominantly composed by hydrophobic residues and is essential for aSyn aggregation and Lewy body formation; the C-terminal region, highly enriched in acidic residues, which has been proposed to contribute to protein stability. In addition, this domain has been implicated in interactions with other proteins and ligands such as DA [34,35] and contains the majority of post-translational modification sites. Several factors have been shown to participate in triggering the aggregation and fibril formation of aSyn, including high protein concentration, mutations and post-translational modifications [36].

Several research papers have highlighted the importance of the mutations in the *LRRK2* gene in synaptic pathology [37,38]. LRRK2 is a large multi-domain protein consisting of a central GTPase Ras-of-Complex (ROC) domain, a COR (C-terminal of Roc) and a kinase domain, surrounded by several protein–protein interaction regions, among which a C-terminal WD40 domain and ARM (armadillo repeat), ANK (ankyrin-like repeat) and LRR (leucine-rich repeat) regions that reside in the N-terminal domain [39].

LRRK2 is extensively expressed through the brain, in the olfactory bulb, striatum, cortex, hippocampus, midbrain, brainstem and cerebellum [40,41]. 

Mutations in LRRK2 cause late-onset, autosomal dominant PD clinically comparable to the idiopathic disease [42,43]. Of the reported pathogenic mutations, the G2019S is the most common, representing 4% of familial and 1% of idiopathic PD worldwide [44]. This substitution leads to a toxic gain of function of LRRK2 kinase activity [45,46]. At the cellular level, LRRK2 is associated with different organelles and structures, such as the Golgi apparatus, endoplasmic reticulum (ER), lysosomes and mitochondria, as well as the microtubule network [40,47]. This ubiquity strengthens the importance of this protein in cell physiology.

Emerging evidence suggests that this large enzyme is also deeply implicated in the regulation of synaptic vesicle dynamics, including neurotransmitter release and receptor mobilization [48]. This is also supported by LRRK2 localization and interaction with multiple presynaptic proteins including N-ethylmaleimide-sensitive fusion protein (NSF) and synapsin 1 [49]. Interestingly, LRRK2 also interacts with Rab5b on synaptic vesicles to modulate endocytic vesicle trafficking [50]. Furthermore, overexpression of both WT and mutant LRRK2 in neurons are associated with a decrease in synaptic vesicle endocytosis and trafficking [51]. More recently, silencing of LRRK2 in mouse cortical neurons affected synaptic vesicle recycling and redistribution, indicating that LRRK2 controls storage and mobilization of vesicle pools inside the presynaptic terminal [52]. Lastly, knock-out of LRRK2 impairs clathrin-mediated synaptic vesicle endocytosis and neurotransmission [48].

Recent studies report that mutant LRRK2 can also affect DA receptor turnover both in cellular and animal models [53,54]. Upon DA treatment, G2019S-LRRK2 caused a strong impairment in DA receptor D1 (DRD1) internalization in SH-SY5Y cell lines. Additionally, in cells expressing the DA receptor D2 (DRD2) mutant LRRK2 decreased the rate of DRD2 trafficking from the Golgi complex to the cell membrane, compared to WT-LRRK2 expressing cells [54].

Glutamate and DA signaling have also been connected in the context of LRRK2 function. Mice carrying the G2019S mutation display increased sensitivity to DRD2 stimulation in the reduction of glutamate transmission onto striatal neurons [55]. In addition, the same group found that DRD2 agonists confer neuroprotection in the same G2019S-LRRK2 knock-in (KI) model challenged with the neurotoxin rotenone, via preservation of intact mitochondrial function [56]. Thus, the neuroprotection might be related to a reduction in glutamate excitotoxicity, given the increase in glutamate transmission operated by G2019S-LRRK2 (see further below in this section).

Alterations in the biology of presynaptic transporters have been studied in humans utilizing positron emission tomography (PET) imaging. Specifically, changes in DA and serotonin transporters (DAT and SERT, respectively) were investigated in idiopathic PD, LRRK2 PD and LRRK2 asymptomatic carriers. Intriguingly, LRRK2 PD patients display similar alterations in DAT and SERT, compared to idiopathic PD, while unaffected LRRK2 mutation carriers have an increase in SERT binding in specific brain regions, such as striatum and hypothalamus [57]. The absence of a healthy control group hampers generalized conclusions, but overall imaging studies suggest that synaptic changes occur in PD and might arise before diagnosis.

PET technique has also been applied to evaluate DAT and VMAT2 (vesicular monoamine transporter) density in 12-month old G2019S-LRRK2 BAC transgenic rats. No abnormalities in DAT, VMAT2 or DA synthesis were observed, and biochemical analyses supported PET findings, as DRD2, DAT and TH levels were comparable between transgenic rats and controls. However, motor performance progressively decreased up to 12 months of age [58]. Of note, transgenic rats display substantial differences in several synaptic and behavioral phenotypes when compared to genetic mouse models of LRRK2, calling for caution when interpreting results from a single animal model.

Remarkably, LRRK2 has also been reported to influence synaptic physiology. Several LRRK2 genetic rodent models have been developed and found to display opposing effects on DA release in WT-LRRK2 transgenic mice and in transgenic rats expressing mutant G2019S-LRRK2, suggesting a presynaptic site of action for this protein [59,60]. The differences in DA phenotypes might be species-specific or related to different transgenesis techniques, but still indicate that LRRK2 acts at the release site of DA neurons. Moreover, the G2019S mutation has been shown to consistently produce aberrant glutamatergic transmission in different KI models. This abnormal transmission is dependent on kinase activity, reversed by pharmacological LRRK2 inhibition and are likely linked to DRD2 activity on presynaptic terminals [37,61,62].

The association between LRRK2 and mitochondria suggests a possible role for this protein in mitochondrial dysfunction leading to PD pathology. Indeed, G2019S-LRRK2 PD patients fibroblasts showed several morphological abnormalities in mitochondria [63], and in SH-SY5Y cells, the overexpression of WT-LRRK2 produced mitochondrial fragmentation exacerbated by G2019S mutation [64]. 

High energy demand sites such as synapses, synaptic vesicle recycling and neurotransmitter release rely on several ATP-consuming steps and on a tight control of cytosolic Ca^2+^ levels, highly depending on functional and healthy mitochondria [65]. This also suggests the importance of a precise control of the clearance dynamics of this organelle. For example, primary mouse cortical neurons expressing G2019S-LRRK2 demonstrated increased mitophagy associated with altered Ca^2+^ levels [66], and ultrastructure examination in G2019S-LRRK2 transgenic mice showed an accumulation of damaged mitochondria, consistent with altered mitophagy in aged mice [67]. These events are contradictory as an increased mitophagy should clear mitochondria more efficiently. However, the studies utilized different models (transfected primary neurons vs. transgenic mice) and different measures (e.g., total mitochondrial content vs. electron microscopy analysis of mitochondrial quality). While it is still difficult to reconcile these findings, it appears that mitochondrial biology is affected in PD conditions and it is likely that at synapses, this could contribute to early synaptic dysfunctions, as previously mentioned [55,56]. In this regard, it is worth noting that LRRK2 kinase activity impairs the interaction between Parkin and Drp1 with consequent defects in mitophagy efficacy [68]. Consistently, mitophagy and Parkin have been reported to play regulatory roles at the synaptic site [69,70]. 

Evidence of structural alteration to neurons and synapses also comes from data from overexpression of mutant LRRK2 in primary cortical neurons, exhibiting neurite injury and retraction that precede neuron loss [71]. Recently, mutant LRRK2 expression in differentiated SH-SY5Y cells has been shown to increase the number of autophagosomes in neuritic-like protrusions [72]. 

Importantly, LRRK2 also modulates the accumulation of pathologic aSyn ([73,74] and our data, Obergasteiger et al., submitted manuscript and BioRxiv doi: 10.1101/707273). All these lines of evidence suggest and support the hypothesis for a key role of LRRK2 and aSyn in modulating synaptic function as an early phenotype in PD. The link between LRRK2 and aSyn to autophagy suggests that this process might also be involved in synaptic alterations, and not only Lewy pathology, in PD pathogenesis [75,76].

## 3. Autophagy-Lysosome Pathway and PD

### 3.1. Autophagic Processes

Autophagy is a catabolic process mostly dedicated to the degradation of cytoplasmic content and organelles, evolutionary well-conserved in eukaryotic cells [77]. While the Ubiquitin-proteasome system (UPS) is mainly implicated in the degradation of short-lived proteins, the ALP is responsible for the breakdown of long-lived proteins, protein aggregates and entire organelles [78]. The distinct types of autophagy differ in the substrate handling, their regulation and conditions in which each of them is preferentially activated, but they all share a common effector, the lysosome [64]. The autophagic processes are classified in macroautophagy, CMA and microautophagy [78]. 

CMA is a highly specific type of autophagy that selectively recognizes cytosolic proteins containing lysosomal targeting motifs (such as KFERQ or VKKDQ), which uses cytosolic chaperones to unfold and translocate them into the lysosome for degradation. At the lysosomal membrane, substrates interact with the lysosome-associated membrane protein 2A (LAMP2A) and cross the membrane to be degraded in the lumen [79,80].

Microautophagy is characterized by the direct sequestration of cytosolic content into the lumen of the lysosome where they are rapidly digested. However, to date, little clarity exists on the exact mechanisms of this type of autophagy.

On the other hand, macroautophagy is the most studied subtype and is generally referred to as autophagy (see Figure 1). It serves two major purposes: providing a source of essential macromolecules (amino acids, lipids, nucleotides) and energy during starvation, and removing dysfunctional or toxic intracellular components. Cytosolic double membranes, called phagophores, engulf intracellular components or organelles generating vesicles termed autophagosomes. These fuse with the lysosome forming the autophagolysosome, in which the cargo content is then degraded. Different organelles are processed by autophagy, including mitochondria (mitophagy), peroxisomes (pexophagy), ribosomes (ribophagy) and parts of the nucleus (nucleophagy) [79,80]. In addition, one of these specializations targets protein aggregates (aggrephagy) with direct relevance to neurodegeneration [81].

Autophagy is regulated by the Atg family of proteins (AuTophaGy-related), which participates to each different step. Notably, over 15 Atg proteins as well as phosphatidylinositol 3-kinase type I and III (PI3-Ks) are involved in the formation and maturation of the autophagosome. 

The initiation of autophagy can be induced by two protein complexes. The first one, PI3-K type III, is composed of Beclin1, Atg14L, VPS34, VPS15 and Ambra1. It helps promoting the generation of omegasomes, which are the initial membrane components of the isolation membrane [82]. These supply the pre-autophagosomal structure adjusting two ubiquitin-like conjugation systems (consisting of Atg12, Atg5, Atg16 and LC3B-II). Progressively, these systems elongate the phagophore membrane and form the mature autophagosome [83]. 

The second is the ULK1 (Unc51-like kinase) complex, which is composed of Ulk1, FIP200, Atg13 and Atg101; when de-phosphorylated, it initiates autophagy through phosphorylation of multiple substrates. As mentioned above, there are two conjugation reactions regulated by Atg7 required for autophagosome biogenesis. First, the ubiquitin-activating enzyme E1-like protein Atg7, induces the formation of the Atg5-Atg12 complex. Then, the conjugation of phosphatidylethanolamine (PE) to LC3B forms LC3B-II, which translocates from the cytosol to the autophagic membrane. LC3B-II is then deconjugated and released from the membrane [76]. After its formation and before fusing with lysosomes, the autophagosome can combine with different endocytic compartments, such as early and late endosomes or multivesicular bodies [84,85]. Once completed, autophagosomes and their cargo are shuttled to lysosomes for fusion and subsequent degradation. Multiple complexes facilitate autophagosome-lysosome fusion including Rab7, LAMP1 and LAMP2, which are required for the trafficking of lysosomes and for autophagosome-lysosome fusion [86,87,88]. In the late stages of autophagy, the maturation of autophagosomes, their fusion with endosomes and lysosomes, the acidification of lysosomes and the recycling of metabolites are further regulated by additional molecules [89].

### 3.2. Autophagy-Lysosome Pathway in PD Etiology: Impact of PD-Linked Genes

Neuronal levels of aSyn and efficient aSyn degradation through autophagic pathways are one of the key elements in PD pathogenesis [90]. Figure 1 summarizes the major roles of LRRK2 and aSyn in the regulation of autophagy, with a specific focus for the presynaptic site. 

Several studies have shown that aSyn-containing inclusions reduce autophagic activity during autophagosome maturation and their fusion with lysosomes [86]; aSyn itself reduces autophagy by promoting the mislocalization of Atg9 [19,91].

As mentioned before, LRRK2 mutations are the most common genetic cause of PD with G2019S increasing its kinase activity and described to impair autophagic processes, leading to a significant accumulation of aSyn in vitro and in vivo [92,93,94]. Recently, it has been clarified that LRRK2 phosphorylates specific Rab proteins (Rab8a, Rab10) with a role in intracellular vesicle trafficking and accumulating studies suggest that ALP dysfunction may impact on LRRK2-associated PD [95,96]. Indeed Rab29, a genomic risk factor for PD, recruits LRRK2 to stressed lysosomes, where it then phosphorylates Rabs to maintain lysosomal homeostasis [97]. Other studies indicate that interactions between Rab29 and LRRK2 could affect ALP function, thus supporting the promising role of this specific pathway in PD neurotoxicity [98]. In addition, pathogenic LRRK2 decreases the activity of Rab7, causing an impairment in the maturation of late endosomes and a delay in degradative trafficking pathways [99], in accordance with the role of Rab7 in endosomal maturation (see Section 3.1). 

Another PD-related gene, VPS35, may also be involved in ALP [100]. Mutations in VPS35 cause autosomal-dominant PD, with the D620N mutation increasing the phosphorylation of Rabs via LRRK2. WT-VPS35, but not mutant, rescues endolysosomal defects caused by mutations in LRRK2 and Rab29 [101]. Moreover, mutations in VPS35 may impair macroautophagy and CMA [102,103].

Loss of function mutations in the *ATP13A2* gene cause Kufor-Rakeb syndrome, characterized by early-onset parkinsonism. This gene encodes the homonymous lysosomal ATPase that is essential for the maintenance of lysosomal pH and autophagosome-lysosome fusion. Mutations in ATP13A2 could impair this process, leading to accumulation of autophagosomes, increasing aSyn concentration and promoting its aggregation, thus worsening neurotoxicity [19,104].

We previously mentioned that Parkin contributes to mitophagy (Section 2). Recently, fibroblasts from Parkin PD patients have been shown to display defective lysosomal function, impaired autophagic flux and several abnormalities in mitochondrial biology [105]. Thus, it is plausible to hypothesize that PD-linked proteins form a complex network that regulates ALP at different levels, with several intersections between players. This complexity is consistent with a common biological substrate (i.e., intracellular vesicle trafficking) in the pathogenesis of PD [9].

## 4. Autophagy-Lysosome Pathway and Proteinopathy

### 4.1. LRRK2 and aSyn Accumulation

LRRK2 and aSyn are key players in PD, as they appear to have central roles in PD pathogenesis and neuropathology. It is now well accepted that LRRK2 dysfunction influences aSyn accumulation, pathology and toxicity through its kinase activity. This occurs through alteration of fundamental cellular functions and signaling pathways including ALP, vesicle trafficking and synaptic homeostasis.

One of the seminal reports on the involvement of LRRK2 in autophagy and consequent modulation of aSyn accumulation comes from the work of Orenstein et al. [94]. Here, they showed that LRRK2 is normally degraded by CMA, whereas the overexpression of both G2019S- and WT-LRRK2 reduced the functioning of this pathway through enhanced binding to LAMP-2A. This blocks the receptor complex formation and prevents the translocation of substrates into the lysosomal lumen for degradation, inducing a detrimental effect on aggregation-prone proteins such as aSyn. This protein then tends to oligomerize into toxic species. Interestingly, it has been shown that WT-LRRK2 competes for the association to lysosomes with monomeric aSyn, as well as with other autophagy substrates. The increasing concentration of WT-LRRK2 decreased aSyn oligomers in lysosomes by competitive binding with monomers, whereas the combination of WT-LRRK2 with A53T-aSyn markedly increased the formation of oligomers in a dose-dependent manner. These findings suggest and support that the concomitant presence of WT-LRRK2 and mutant aSyn at the lysosomal membrane exacerbates autophagic defects that can eventually trigger and sustain aSyn cytotoxicity. 

This is consistent with positive effects of LRRK2 silencing or kinase inhibition on neuropathology induced by recombinant aSyn fibrils [73,74]. Our unpublished results (manuscript under review and Obergasteiger et al., 2019, BioRxiv, doi: 10.1101/707273) suggest that not only CMA is affected by LRRK2 kinase activity, but also the formation of autolysosomes is enhanced by pharmacological kinase inhibition, leading to enhanced clearance of endogenous pS129-aSyn. Consistently, LRRK2 has been reported to impact several steps of the ALP [106,107]. 

Previously, G2019S-LRRK2 SH-SY5Y cells were shown to exhibit neurite shortening in concomitance to a significant increase in LC3B-positive autophagic vacuoles in both somatic and neuritic compartments, upon differentiation with retinoic acid. The pharmacologic stimulation of autophagy with rapamycin promoted the increase in neuritic autophagic vacuoles and neuritic shortening observed in G2019S-LRRK2 cells. In comparison, the knockdown of LC3B or Atg7 reversed the effects of G2019S-LRRK2 expression on process length, suggesting an association between LRRK2 mutation and autophagy on neurite remodeling [72]. It is worth noting here that these results point to a stimulatory effect of the G2019S mutation on autophagy, while other reports indicate the opposite. This contradiction on the specific direction of effect has been previously discussed with regard to mitochondrial biology (Section 2). We discussed this currently unresolved conundrum in a previous review article [108], to which we direct the reader for deeper insights. Nevertheless, we need to acknowledge that this is an outstanding question in the field. Numerically speaking, there appears to be more studies finding an inhibitory role for G2019S-LRRK2 in ALP, but a significant number of investigations demonstrating the opposite are present and these cannot be disregarded.

Using human genomic DNA expression systems, LRRK2 was detected at multivesicular bodies and autophagic vacuoles in Vero and HEK293 cells, and in the human brain. Specifically, LRRK2 was found to localize to specific membrane subdomains, including the neck of caveolae, microvilli and endosomal–autophagic structures. Interestingly, the expression of R1441C-LRRK2 induced an impairment of autophagy with the accumulation of multivesicular bodies and abnormally enlarged autophagic vacuoles together with increased levels of p62 and skein-like cellular lesions. On the other hand, LRRK2 knockdown increased autophagic activity preventing bafilomycin-induced cell death under starvation conditions [92].

More recently, the functional relationship between LRRK2 and ALP was investigated in primary neurons expressing the human G2019S or R1441C mutations [109]. The expression of G2019S- or WT-LRRK2 inhibited autophagosome production, whereas R1441C-LRRK2 decreased the autophagosome/lysosome fusion and worsened lysosome-mediated cargo degradation. This was also confirmed by in vivo analysis of cortex and SNpc from 22-month-old R1441C-LRRK2 transgenic animals in which alterations in autophagosome number and increased LC3B levels were reported. Interestingly, MLi-2 and PF-06447475 kinase inhibitor treatment did not revert the alterations in autolysosome maturation, indicating a lack of dependence from LRRK2 kinase activity in this specific model. Lastly, this study showed that both WT- and G2019S-LRRK2 bind to the a1 subunit of the vacuolar ATPase (that mediates lysosome acidification), while this is abolished by the R1441C mutation, leading to a decrease in a1 protein and cellular mislocalization.

In addition to several in vitro studies, the function of LRRK2 in autophagy and consequences on aSyn processing have been extensively investigated in animal models. A thorough overview of these studies can be found in a recent review [110]. Here, we will provide a brief summary of evidence that strengthens the connection between these processes and aSyn accumulation. 

The transgenic expression of G2019S-LRRK2 in mice induced progressive degeneration of nigrostriatal DA neurons and reduced DA neurite complexity in primary cultures. Moreover, these findings were paired with an increase in the size of autophagic vacuoles, in autophagosome number and aggregated damaged mitochondria in mouse striatum [67]. 

In addition to transgenic models, KI mice have also been developed, which provide a more accurate etiologic modeling of LRRK2 PD. 

G2019S KI mice displayed an age-dependent decrease in basal and evoked striatal DA release in the striatum, which was matched by severe mitochondrial abnormalities and increased LC3B-II levels in older G2019S mice [111]. 

These data were replicated in primary cultured neurons, where G2019S-LRRK2 deeply impacts lysosome morphology and function, decreasing lysosomal protein expression and size. Cortical neurons from G2019S-LRRK2 KI mice show alkalinized lysosomes and decreased autophagic flux; specifically, the levels of LAMP1 and LC3B-I were both significantly decreased in adult G2019S animals compared to WT controls. Moreover, these changes were accompanied by an accumulation of endogenous, detergent-insoluble aSyn and increased release of neuronal aSyn. These effects were linked to kinase activity as the LRRK2 inhibitors GSK2578215A and CZC-25146 decreased the adverse effects on aSyn and rescued the lysosomal deficits [112]. This evidence suggests that PD-associated mutations in LRRK2 alter aSyn clearance through lysosomal dysfunction.

Using a well-established model of synucleinopathy [113], two additional studies showed that LRRK2 influences formation of aSyn aggregates both in vitro and in vivo, and that the G2019S mutation increases sensitivity towards pathology and accelerates the progression of aSyn inclusion formation.

In cultured neurons and in the rat DA SNpc, G2019S-LRRK2 BAC transgenesis enhances endogenous aSyn recruitment to inclusions and exacerbates the effect of aSyn pre-formed fibrils (PFFs). The knockdown of aSyn with antisense oligonucleotides decreased the formation of PFF-induced inclusions, both in G2019S-LRRK2 expressing neurons and controls, indicating that endogenous aSyn is required to form inclusions independently from LRRK2 mutation status. Importantly, LRRK2 kinase inhibitors (i.e., PF-06447475 and MLi-2) strongly reduced the development of aSyn pathology in mutant and control neurons, demonstrating that the effects of aSyn PFFs are mediated by LRRK2 kinase activity [73].

In addition, LRRK2 antisense oligonucleotide treatment reduced pathologic aSyn inclusions, DA cell loss and associated motor deficits in PFF-treated WT mice, highlighting a role for endogenous, non-mutant LRRK2 in induced synucleinopathy [74]. Both these therapeutic approaches are considered a potential application for novel clinical neuroprotection strategies, not only for LRRK2 mutations carriers but also for idiopathic PD patients.

Altogether, these data suggest that LRRK2 may function upstream of aSyn in modulating its aggregation and cytotoxicity in neurons, via modulation of protein degradation pathways [43]. Indeed, several LRRK2-PD patient brains display Lewy body pathology but also Tau pathology is abundantly represented [114], increasing the clinical relevance of these biological processes. However, it must be mentioned that the same group reported no effect of LRRK2 kinase inhibition on induced neuropathology in rodents [115]. Future studies will need to address these discrepancies, possibly through the harmonization of models and paradigms employed.

### 4.2. Targeting the Autophagy-Lysosome Pathway to Reduce aSyn Accumulation

The findings that defective autophagy has an established role on aSyn toxicity has provided a rationale for possible therapeutic interventions targeting ALP. To identify at which point the autophagic machinery fails and which are the best approaches to restore neuronal and synaptic proteostasis is vital and essential for therapy development. Among the different approaches aimed at restoring the autophagic machinery, the application of rapamycin as an autophagy inducer is a common one. Rapamycin is an mTORC1 inhibitor and has been demonstrated to prevent protein aggregation and neurodegeneration in experimental models of PD. Interestingly, in mice with intact autophagy, mTORC1 inhibition with rapamycin acutely increased autophagic vacuoles formation in axons, decreased the number of synaptic vesicles and depressed evoked DA release [116]. These results point to a synaptic function of mTORC1-mediated autophagy in DA neurons, aside from proteinopathy.

Intriguing results have been obtained in several in vitro and in vivo models of PD using trehalose, a disaccharide that induces autophagy in a mTORC1-independent fashion and is able to regulate TFEB activity, a master regulator of lysosome biogenesis and function. Treatment of H4 cells and primary neurons with trehalose, before exposure to aSyn, prevents lysosomal enlargement and aSyn aggregate formation [117,118]. In A53T-aSyn overexpressing rats, oral administration of trehalose significantly reduced DA neurodegeneration, aSyn accumulation and aggregation in the nigrostriatal system [119]. Recently, chronic treatment of aged G2019S-LRRK2 BAC transgenic mice with trehalose ameliorated deficits in motor and cognitive functions (Pischedda et al., 2019, BioRxiv, doi: https://doi.org/10.1101/721266).

Similar neuroprotective effects were observed in MPTP-treated mice [120], and recently, trehalose has been effectively tested both in rat and non-human primates expressing A53T-aSyn [121]. 

An alternative therapeutic approach aimed at increasing selectivity is based on genetic manipulations to activate and/or overexpress specific autophagic components. For example, overexpression of Beclin1 reduces aSyn aggregation and neurodegeneration in a mouse model of PD [122]. Similarly, overexpression of LAMP2 in rats and cellular PD models restored autophagy, reduced aSyn aggregation and increased the viability of SNpc neurons as well as the density of their axon terminals in the striatum [123]. 

Lastly, the natural product curcumin has been shown to induce autophagy, to protect from DA neurodegeneration induced by rotenone, and accelerate the elimination of A53T-aSyn through inhibition of mTORC1 [124,125]. 

Collectively, these studies support ALP as a promising therapeutic target to combat proteinopathy. However, it is currently not clear how this evidence could be exploited in a clinical setting. In addition to the controversial role of neuropathology in neuron cell death [126], the timing of these therapies would be complicated as they will be likely applicable only at the late stages, when the vast majority of neurons is already lost and slowing the disease would be improbable. For this reason, targeting early dysfunction directly linked to pathogenic processes would be a more effective strategy. We posit that elucidating the mechanisms involving LRRK2, autophagy and early synaptic function has the potential to inform those strategies.

## 5. Autophagy-Lysosome Pathway and Synaptic Transmission

As mentioned above, the essential role of autophagy is to provide energy and eliminate toxic cellular components such as protein aggregates, damaged organelles or pathogens. In addition, it is implicated in the regulation of a wide range of physiological and pathological processes, including responses to starvation, intracellular clearance, development, immunity, aging, cell death and survival and lastly synaptic maintenance and communication [77]. Therefore, it is not surprising that the dysregulation of this process has considerable consequences and might be implicated in early synaptic dysfunction in aging and neurodegenerative diseases, such as PD. 

Genetic studies have indeed demonstrated that in vivo conditional ablation of core autophagy genes, such as Atg5 or Atg7, leads to the accumulation of aggregates containing ubiquitinated proteins and to progressive neurodegeneration, indicating that this catabolic pathway is required for neuronal and synaptic maintenance [127,128].

Impairment in synaptic structure and function ultimately cause axonal disruption and neurodegeneration. Synaptic autophagy can thus be considered as the main connection between synaptic dysfunction and neurodegeneration, with several canonical synaptic proteins, including LRRK2 and aSyn, playing important roles in autophagic processes at synapses. In addition, signaling pathways, including Endophilin A and synaptojanin1 (Synj1), are also being clarified [11]. Evidence on the involvement of LRRK2 and aSyn in synaptic transmission is presented earlier. Below we discuss the roles of additional molecular players that might link these PD-proteins to autophagy at the synaptic site.

### 5.1. Endophilin A

Endophilin A was recently identified as a novel risk factor for PD [129]; it is a core protein for synaptic vesicle biogenesis that interacts with Synj1 and dynamin [130]. In Drosophila models, the phosphorylation of Endophilin A is enhanced by G2019S-LRRK2 expression, which biases its membrane-deforming capabilities and the formation of high curvature membranes that recruit Atg3 and thus initiate autophagosome formation [131,132]. 

Interestingly, Endophilin A colocalizes with Atg5 and LC3B in autophagosomes, interacts with Beclin1 and regulates the formation of autophagosomes [133]. Specifically, the Bar domain of Endophilin A, responsible for membrane binding, provides the driving force for the formation of autophagosomes [134].

Recent analyses revealed changes in pathways including synaptic transmission and protein homeostasis. Specifically, increased FBXO32 (a Ubiquitin E3 ligase) directly interacted with Endophilin A and autophagosomes. Mice expressing mutant Endophilin A display upregulated FBXO32 expression that colocalizes with Endophilin A and is required for autophagosome formation [135]. Lastly, Endophilin A phosphorylated by LRRK2 provides a docking station at nerve terminals for autophagy proteins, including Atg3 [136].

### 5.2. Synaptojanin1 (Synj1)

Synj1 binds Endophilin A and regulates phosphoinositide content in membranes during the endocytosis of synaptic vesicles [137,138]; in addition, mutations in its coding gene (*SYNJ1*) have been identified to cause early-onset PD [139].

Recent studies found that Synj1 participates in autophagy at the presynaptic terminal. In particular, the PD-linked mutation R258Q produces an imbalance in lipid production with consequent accumulation of WIPI2/Atg18a, which then blocks autophagosome maturation [140]. 

Furthermore, Synj1 also plays a role in the trafficking of synaptic proteins into endolysosomes, with accumulation of large vesicular structures, enlarged acidic vesicles, abnormal endosomes and impaired autophagy [141]. Collectively, these findings indicate that Synj1 PD-linked mutation affects vesicle trafficking and protein turnover at synapses, supporting the overall hypothesis of disruption in synapse-specific autophagy [10,142].

### 5.3. Additional Regulatory Proteins and Interactions with LRRK2

Dynamin is a critical GTPase mediating membrane scission during endocytosis, including the recycling of synaptic vesicles [143]. Genetic and biochemical interactions between dynamin 1–3 and LRRK2 have been recently uncovered. In particular, LRRK2 interacts with dynamin 1 in neuronal cultures [144]. Furthermore, genomic variability in *DNM3* (coding for dynamin 3) modifies the age of onset in G2019S-LRRK2 PD patients and recently dynamin 2 has been shown to participate in the activation of mTORC1 [145,146]. 

Bassoon is a scaffold protein involved in the structural organization of the presynaptic cytoskeleton that regulates synaptic transmission and synaptic autophagy. Indeed, Bassoon binds to Atg5, and synapses lacking both Piccolo and Bassoon accumulate autophagosomes [147].

The small GTPase Rab26 is also related to synaptic autophagy. Overexpression of active Rab26 induces vesicle clustering in the neuronal body. These clusters colocalize with autophagy proteins such as Atg16L1 and LC3B [148].

Synaptobrevin, also known as VAMP2, is a core component of the synaptic vesicle exocytosis machinery and has been proposed to regulate endolysosomal degradation. Deletion of synaptobrevin produces the accumulation of endosomes, impairment of protein degradation and increased autophagy [149].

The implication of many of these proteins in synaptic homeostasis suggests dysfunction of the ALP as a key contributor to synaptic abnormalities and PD pathogenesis (Figure 1). Studies in PD animal models, but also other neurodegenerative diseases, suggest that changes in synapses and axons represent the earliest detectable phenotype [71,72]. Interestingly, R1441G-LRRK2 transgenic mice display axonal pathology in the nigrostriatal pathway at an early age [150] before more severe phenotypes appear [59]. Additionally, functional imaging in PD patients show that early changes in DA terminals in the striatum can occur many years before SNpc cell loss [151]. These findings indicate that the detrimental changes taking place in axon terminals following defects in autophagy and synaptic vesicle trafficking could likely represent the primary event of degeneration, subsequently leading to retrograde degeneration of nigral neurons [142].

## 6. Autophagy-Lysosome Pathway Links Early Dysfunction to Late Pathology to Provide Disease Modification

Traditionally, synaptic function and proteinopathy have been separate fields of studies in the larger theme of neurodegeneration research. The molecular machineries implicated have been somewhat distinct until recent years, when elucidation of clearer intracellular vesicle trafficking processes highlighted several points of convergence. In particular, autophagy has been demonstrated in the recent past to play local functional roles at presynaptic sites, distinct from its catabolic function. Interestingly, it is being better appreciated that lysosomes also act beyond their degradative capacities but participate in cellular signaling, for example regulating local Ca^2+^ microdomains [152]. This forms the basis of our hypothesis that modulation of ALP might be the missing temporal link between early synaptic failure and later neurodegeneration/neuropathology.

Indeed, the autophagic machinery decreases in efficiency with aging and during disease. This has been linked to accumulation of aggregation-prone proteins, such as aSyn. However, this dysregulation could trigger earlier deficits at the synapse, impairing synaptic dysfunction as initial events in PD neurodegeneration. Consistently, LRRK2 and aSyn, the most studied PD genes, are strongly implicated both in ALP and synaptic regulation, as has been discussed. In addition, in some cases, synaptic alterations due to a defective protein could compromise normal autophagy, reinforcing the link between these players in the pathogenic process.

Clarifying the exact mechanisms and which process “comes first” in the cascade of dysfunctional events is the goal of future studies. 

As synaptic dysfunction is an early manifestation of an underlying pathology, ALP could be a valid therapeutic strategy for targeting the disease at an early stage. This would be at a juncture where neuronal loss had not yet occurred at a substantial level [11], and reversal of the dramatic protein aggregation characteristic of the final disease stages was still possible. Our knowledge of the molecular mechanisms involving LRRK2 and aSyn could be leveraged to clarify these processes and nominate novel targets in these pathways.

We thus postulate that targeting ALP early might prevent later neurodegeneration in PD. In order to make this applicable to clinical practice, novel diagnostic procedures will be required that will permit the assessment of synaptic function in non-symptomatic patients. To aid this, pre-symptomatic carriers of PD-linked mutations will likely play a crucial role in helping the identification of early dysfunction, as it has been reported for LRRK2 mutation carriers showing deficits in functional network connectivity [153]. 

Future work should then be addressed at providing a direct correlation between synaptic dysfunction and neuropathology. For example, an ideal candidate for therapeutic strategy would be a protein/molecule involved in the ALP pathway that is also capable of modulating primary synaptic function and later handling of aSyn. LRRK2 seems to meet these prerequisites and clinical trials with kinase inhibitors are ongoing. Nevertheless, identifying additional players in a LRRK2 pathway will provide additional options for therapy.

## Figures and Tables

**Figure 1 cells-09-01115-f001:**
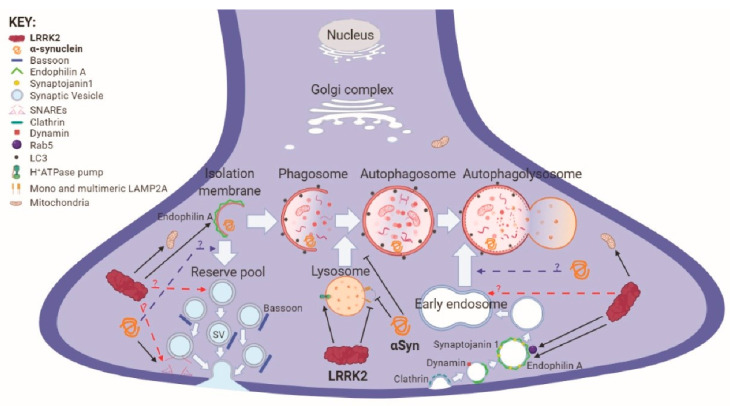
*LRRK2 and aSyn regulation of autophagy at the presynaptic terminal*. Macroautophagy at the synaptic terminal is not only implicated in degradation of cytosolic content but can also modulate vesicle release and endocytosis, with many common regulators. In addition, PD-linked proteins act on both mechanisms, likely modulating a common process. For example, LRRK2 and aSyn regulate autophagic flux and lysosomal activity, with PD-causing mutations impairing the process. Nevertheless, LRRK2 phosphorylates Endophilin A with effects on synaptic endocytosis, which could in turn impact the endolysosomal system. A similar link might be possible in the regulation of synaptic vesicle release, through interactions with Bassoon and aSyn. The latter, in addition, is implicated in several steps of the ALP, in presynaptic vesicle release and the fusion of endosomes with lysosomes. Thus, there are several points of convergence between autophagy and synaptic function, with LRRK2 and aSyn playing critical regulatory roles.

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
