# Peer review of "Alpha-Synuclein and LRRK2 in Synaptic Autophagy: Linking Early Dysfunction to Late-Stage Pathology in Parkinson’s Disease"

_cells, 2020, doi:10.3390/cells9051115_

Round 1
Reviewer 1 Report
The manuscript is a very thorough review of the role that autophagy plays in the context of normal synaptic transmission, in a-synuclein accumulation, and as a contributor to the pathology seen in Parkinson's disease. This is a subject that has been coming up with increasing frequency in the Parkinson's disease field, but there are few systematic reviews of the topic making the content timely as well as important and of wide interest. The major strength of the review lies with its thorough and scholarly coverage. There are some minor issues that if addressed would improve synthesis and readability.
- While a key strength lies with the review's thoroughness, it is sometimes encyclopedic--invoking a list of items having equal weights, and could be improved by increased synthesis. Some of this may have been due to a formatting error since there are several paragraphs that appeared in the transcript as single sentences (e.g. lines 139-144 and lines 145 and 147), but in other cases, some of the data are summarized very succinctly in a way that makes it hard to pull the data together and/or know where the strengths and weaknesses sit. For example, line 193 discusses a G2019S associated increase in mitophagy and line 194 cites an accumulation of damaged mitochondria in G2019S transgenic mice. These two findings on the surface seem contradictory--an increase in mitophagy should clear the damaged mitochondria, so it is not clear how the facts fit together. And there are several lines of evidence suggesting that G2019S generally inhibits autophagy, but other evidence (e.g. line 193, 309), supports a pro-autophagy role. Clearly there are experimental and contextual differences, but a line or two here and there that either directly acknowledge a conundrum or serves to define the contexts would be helpful for the reader.
- Line 167: It is not clear why the SERT binding pattern in unaffected LRRK2 mutation carriers might indicate early PD-related changes in the absence of a normal control population.
- A figure citation does not appear in the text, and it would be very useful to have the figure appear much earlier in the review, beginning with the description of the autophagic process in part 3. The font size in the key and near the release site is also too small.
- Line 152, Is a synaptic vesicle still called a synaptic vesicle on the endocytic side? Or is Rab5b tagging endosomes?
- Line 211, "demolishment" should be "demolition".
- Line 400, "abilities" should probably be "abnormalities".
Reviewer 2 Report
The article entitled “LRRK2, alpha-synuclein and synaptic autophagy: linking early dysfunction to late-stage pathology in Parkinson´s disease” by Lamonaca and Volta is a review covering aspects of alpha-synuclein and LRRK2 patho-physiology in synaptic processes in relation to autophagic protein turnover in Parkinson’s disease. The manuscript is well organized and provides a very good overview on the topic, with an original perspective. Below, some suggestions that may help improving the quality of the manuscript.
Major points
1) Description of alpha-syn physiology in synapses and its pathological changes appear to be fragmented in the first part of manuscript. I suggest to put together the description of alpha-syn synaptic physiology in a paragraph preceding that dealing with the pathological role of the protein.
2) Chapter 2, first part: The possible post-synaptic alpha-syn effects in PD should also be mentioned. For example, alpha-syn oligomerization may contribute to both physiological and pathological roles of the protein (see Sulzer and Edwards, J Neurochem., 2019). Moreover, pathological implications have been proposed for defective postsynaptic alpha-syn-NMDAR interactions in basal ganglia (see Durante et al., Brain, 2019) or the hippocampus (see Sweet et al., J Neurosci., 2015; Teravskis et al., J Neurosci, 2018) affecting synaptic plasticity.
3) Chapter 2, second part. The role of the G2019S LRRK2 mutation on aberrant glutamatergic transmission, dopamine transmission and turnover, as well as mitochondrial sensitivity, had been considered in LRRK2 models. It may be relevant discussing these data also in relation of the altered D2R-mediated striatal glutamatergic transmission reported in a G2019S KI mouse model by Tozzi and colleagues (Neurobiol. Dis., 2018 and Cell Death Dis., 2018). Of note, this group also reported in their LRRK2 model reduced striatal dopamine release and increased mitochondrial sensitivity to parkinsonian toxins; this may help the general discussion.
Minor points
- I would change the title as “Alpha-synuclein and LRRK2 in synaptic autophagy: linking early dysfunction to late-stage pathology in Parkinson´s disease”, since the role of a-syn is described first in the manuscript.
- The text should be double-checked for grammar or typos errors (i.e. lines 145 and 180, “showed” (use “shown”); line 184, “suggest”; line 209, “ad organelles”).
- Lines 100-102, “…long-term potentiation (LTP) and long-term depression (LTD) are altered in MSNs 100 from 6-OHDA animals… [17]”. The seminal work of the same group (Calabresi et al., Brain, 1993) might be cited instead.
Reviewer 3 Report
The review is very clear and well written. The message of this paper is very relevant, but I would suggest to authors to deep some aspect.
In particular, the autophagy is too described regardless of the mitophagy, which is very important in PD. To this purpose, the authors forget that LRRK2 influences Parkin protein (PMID:30629163), which is among the key actors of PD. Moreover, it is described also effect of RAB7 on LRRK2 activity (PMID: 25080504) and in this review RAB7 is not mentioned. To this regard, there is also an imprecision at lane 253: RAB7 has a pivotal role in fusion between autophagosome and lysosome (PMID: 15340014)
Finally, they described in paragraph 3.2 the impact of PD-linked genes on autophagic pathway. Recently, it was described effect of PARK2 mutation on autophagic flux and mitochondrial function in PD’s fibroblasts (PMID:31091796)
